# Crosstalk between Schizophrenia and Metabolic Syndrome: The Role of Oxytocinergic Dysfunction

**DOI:** 10.3390/ijms23137092

**Published:** 2022-06-25

**Authors:** Kah Kheng Goh, Cynthia Yi-An Chen, Tzu-Hua Wu, Chun-Hsin Chen, Mong-Liang Lu

**Affiliations:** 1Department of Psychiatry, Wan-Fang Hospital, Taipei Medical University, Taipei 116, Taiwan; havicson@gmail.com (K.K.G.); cyn.c@icloud.com (C.Y.-A.C.); chunhsin57@gmail.com (C.-H.C.); 2Psychiatric Research Center, Wan-Fang Hospital, Taipei Medical University, Taipei 116, Taiwan; thwu@tmu.edu.tw; 3Department of Psychiatry, School of Medicine, College of Medicine, Taipei Medical University, Taipei 110, Taiwan; 4Department of Clinical Pharmacy, School of Pharmacy, College of Pharmacy, Taipei Medical University, Taipei 110, Taiwan

**Keywords:** oxytocin, schizophrenia, metabolic syndrome

## Abstract

The high prevalence of metabolic syndrome in persons with schizophrenia has spurred investigational efforts to study the mechanism beneath its pathophysiology. Early psychosis dysfunction is present across multiple organ systems. On this account, schizophrenia may be a multisystem disorder in which one organ system is predominantly affected and where other organ systems are also concurrently involved. Growing evidence of the overlapping neurobiological profiles of metabolic risk factors and psychiatric symptoms, such as an association with cognitive dysfunction, altered autonomic nervous system regulation, desynchrony in the resting-state default mode network, and shared genetic liability, suggest that metabolic syndrome and schizophrenia are connected via common pathways that are central to schizophrenia pathogenesis, which may be underpinned by oxytocin system dysfunction. Oxytocin, a hormone that involves in the mechanisms of food intake and metabolic homeostasis, may partly explain this piece of the puzzle in the mechanism underlying this association. Given its prosocial and anorexigenic properties, oxytocin has been administered intranasally to investigate its therapeutic potential in schizophrenia and obesity. Although the pathophysiology and mechanisms of oxytocinergic dysfunction in metabolic syndrome and schizophrenia are both complex and it is still too early to draw a conclusion upon, oxytocinergic dysfunction may yield a new mechanistic insight into schizophrenia pathogenesis and treatment.

## 1. Schizophrenia: A Complex Disorder

Schizophrenia is a debilitating neuropsychiatric disorder that is characterized by a constellation of signs and symptoms of unknown etiology, particularly entailing but not restricted to hallucinations, delusions, disorganized speech, disorganized behaviors, and negative symptoms [1]. The mainstays of schizophrenia treatment remain antipsychotics, namely dopamine receptor antagonists. Although having a lifetime prevalence of 1% [2,3], the diagnosis of schizophrenia has tremendous negative consequences, including unemployment, social isolation, stigma, and early mortality [4]. Schizophrenia has been discussed as a disorder of gene–environment interactions that involve multiple vulnerability factors, including but not limited to genetic risk factors, birth and early life environmental risk factors, psychosocial stressors, psychological factors, gut microbiota, and neurochemical disturbances [5,6]. Given that schizophrenia tends to be a brain disorder with the presence of brain functional abnormalities and neurochemical disturbances, it has been considered that at least two neurotransmitters have emerged as leading contenders for its neuropathophysiology: dopamine and glutamate [1]. Hypofunctional N-methyl-D-aspartate receptors on γ-aminobutyric acid interneurons in the prefrontal cortex that lead to the overactivation or downregulation of glutamate signaling to the ventral tegmental area may in turn result in excess dopamine in the ventral striatum via the mesolimbic pathway. Evidence of an increased dopamine synthesis capacity or the hyperactivation of the dopaminergic mesolimbic pathway has been thought to be associated with positive symptoms of schizophrenia [7]. Beyond that, several other neurotransmitters and neuropeptides, including oxytocin, serotonin, cortisol, GABA, cannabinoid, neuropeptide Y, neurotensin, cholecystokinin, corticotropin-releasing factor, and orexin, have been implicated in the neuropathophysiology of schizophrenia [8,9].

Compared with the general population, persons with schizophrenia are associated with a standardized mortality hazard ratio of 3.1 [10] and a reduced life expectancy by a weighted average of 14.5 years [11]. Attaining full remission of symptoms is one of the greatest challenges in persons with schizophrenia, although they have shown a high degree of stability in their disease state, and their symptoms tend to remain stable over time [12]. A large improvement in years of potential life loss for persons with schizophrenia with respect to suicide and accidents was found over the last two decades. However, this improvement was offset by an increasing number of life-years lost in deaths from preventable chronic physical diseases [13,14]. The introduction of antipsychotics has improved the symptoms and decreased the repeated hospitalizations of persons with schizophrenia. However, the benefits are obscured by their unpleasant side effects, particularly weight gain and metabolic disturbances [15]. Antipsychotics are generally believed to be associated with an increased risk of physical illnesses, including obesity, dyslipidemia, diabetes mellitus, and cardiovascular disease [16,17]. However, a recent nationwide cohort study on persons with schizophrenia suggests that long-term antipsychotics use does not increase severe physical morbidity leading to hospitalization and mortality [18]. Furthermore, studies on cardiovascular and metabolic health in persons with schizophrenia, including untreated and antipsychotic-naïve individuals [19,20], showed clear evidence of an increase in the risk of all cardiovascular diseases, diabetes mellitus, and metabolic syndrome [21,22,23]. This evidence suggests that the risk factors for cardiovascular and metabolic diseases may point to other influences independent of common lifestyle factors or antipsychotic use [24].

## 2. Metabolic Syndrome: Definition, Prevalence, and Pathophysiology

Metabolic syndrome is a condition that is characterized by a cluster of several physical diseases, which together increase the vulnerability of a person to developing cardiovascular disease, diabetes mellitus, and vascular and neurological complications such as cerebrovascular accidents. The World Health Organization (WHO) Consultation proposal [25] was the first attempt to define metabolic syndrome, with the notion that modification is needed according to new research information and the predictive power of the proposed criteria. Subsequently, the criteria of metabolic syndrome are modified according to the proposed formulated definitions of different committees, including the European Group for the Study of Insulin Resistance (EGIR) [26], the National Cholesterol Education Program Adult Treatment Panel III (NCEP: ATP III) [27], the American Association of Clinical Endocrinology (AACE) [28], the International Diabetes Federation (IDF) [29], and the American Heart Association/National Heart, Lung, and Blood Institute (AHA/NHLBI) [30]. These operant definitions agree on the essential components, including obesity, hypertension, and dyslipidemia, while the differences are summarized in Table 1. Metabolic syndrome is diagnosed when a person fulfills at least three of the proposed variables, including central obesity, increased triglycerides, low high-density lipoprotein cholesterol, increased blood pressure, and increased fasting glucose. As the first proposed definition for metabolic syndrome, the WHO proposal was suited to be a research tool, whereas the NCEP: ATP III definition was more widely adopted in clinical practice [31]. Insulin resistance is acknowledged as being an important causative factor for metabolic syndrome but is limited by its difficulty to measure in clinical practice. Central obesity, represented by waist circumference, is emphasized in the latter definition of metabolic syndrome as it is useful in predicting a high rate of metabolic syndrome [32], though more precise ethnicity-specific cutoff values for waist circumference should be investigated. In addition, recent research has argued that body fat distribution and impaired adipose tissue function, rather than the total fat mass as measured by the body mass index, better predict individual metabolic disturbances [33]. Although the incidence and the prevalence of metabolic syndrome vary based on the different criteria and different populations, the incidence of metabolic syndrome increased from the 1980s to the 2010s among US adults [34], and the prevalence remained stable at approximately 35% in recent years [35]. In other words, as translated from the known prevalence, about one third to one quarter of the world’s population is affected by metabolic syndrome [36].

Despite the high prevalence and heavy burden of the disease, the pathophysiology of metabolic syndrome encompasses multiple intricate mechanisms that have yet to be fully elucidated. The proinflammatory state brought by obesity and insulin resistance remains at the core of the pathogenesis of metabolic syndrome [37]. The inhibition of lipolysis in the adipose tissue of persons with insulin resistance results in the excessive flux of free fatty acids that in turn alter the insulin signaling cascade in different organs (i.e., reducing glucose uptake in skeletal muscle and promoting glucogenesis and lipogenesis in the liver), further creating a vicious cycle for insulin resistance. The failure to compensate for hyperinsulinemia leads to a decrease in insulin levels that is exacerbated by the lipotoxic effect of free fatty acids on the beta cells of the pancreas [38]. A chronic proinflammatory state that results from the alteration of cytokine production and the activation of inflammatory signaling pathways, primarily induced by obesity or secondarily through insulin resistance, is also responsible for metabolic syndrome. Adipose tissue, apart from its thermoregulation and lipid storage function, is considered one of the largest endocrine organs for its ability to synthesize and release adipokines that are involved in a variety of pathophysiological mechanisms, including the regulation of appetite, glucose and lipid metabolism, inflammation, and cardiovascular homeostasis [39]. In the proinflammatory environments of obesity, the upregulation of proinflammatory adipokines (e.g., leptin, chemerin, resistin, visfatin, vaspin, kallistatin, tenascin C, osteopontin, calprotectin, interleukin-1, interleukin-6, interleukin-8, interleukin-32-alpha, and tumor necrosis factor-alpha) and the downregulation of anti-inflammatory adipokines (e.g., adiponectin, omentin, fibroblast growth factor 21, lipocalin 2, secreted frizzled-related protein 5, and interleukin-10) are reported [38,39,40]. Although research on obesity and insulin resistance to the pathophysiology of metabolic syndrome has exponentially increased, other mechanisms have also been discussed, including chronic stress, dysregulation of the hypothalamic–pituitary–adrenal axis, renin–angiotensin–aldosterone system activity, mitochondrial dysfunction, intrinsic tissue glucocorticoid actions, circadian, microbiota, and the dysregulation of hormonal metabolism effects [41,42,43].

## 3. Schizophrenia and Metabolic Syndrome

### 3.1. Vulnerability to Metabolic Syndrome in Persons with Schizophrenia

Vulnerability to metabolic syndrome is an urgent concern in persons with schizophrenia. Metabolic syndrome denotes a clustering of risk factors for cardiovascular disease and acts as a predictor of cardiovascular mortality [44,45]. Cardiovascular disease accounts for the most prevalent cause of premature mortality among persons of all ages (even in young adults) with schizophrenia [46]. Persons with schizophrenia are at a 4.4-fold risk of central obesity, a 2.7-fold risk of hypertriglyceridemia, a 2.4-fold risk of low HDL, a 1.4-fold risk of hypertension, and a 1.99-fold risk of diabetes compared with the general population [47]. The estimated prevalence of metabolic syndrome in persons with schizophrenia is 37–67%, which is a 2- to 3-fold increased risk compared with the general population [47,48]. The high prevalence of metabolic syndrome among persons with schizophrenia is attributed to disease-related characteristics and lifestyle choices [49], such their negative psychopathology interfering with their insights into physical health and resulting in a sedentary lifestyle and a lack of physical exercise [23,50], their unhealthy high-fat, high-calorie fast food dietary habits and poor nutrition, which are a result of less social support and low income [51], limitations in accessibility to general medical care [52], and increased tobacco use as a self-medicative method to alleviate their psychotic symptoms through normalization of default mode network hyperconnectivity by nicotine [53].

Aside from those aforementioned risk factors for metabolic syndrome, the association of antipsychotic treatment with metabolic syndrome has been widely described in the population of schizophrenia. Almost all antipsychotics are associated with metabolic side effects [54], and the prevalence of metabolic syndrome among persons with schizophrenia has steadily increased over continuous treatment with antipsychotics [55]. As central obesity has served as the primary proxy of metabolic syndrome, weight gain is the most widely discussed metabolic parameter in persons with schizophrenia that are treated with antipsychotics. Although the magnitude of negative consequences differs among antipsychotics, nearly all antipsychotics result in weight gain [56,57,58]. Early increases in body weight of more than 5% in the first month are the best predictor of chronic central obesity in persons with schizophrenia who are treated with antipsychotics [59], as they might continue to gain weight for several years [60]. Metabolic disturbances often develop during the early stages of treatment, and the emergence of these conditions is not strictly related to or even triggered before the onset of the weight gain induced by the antipsychotics [61]. A debate is continuing regarding the independence of other metabolic parameters of antipsychotics-induced weight gain [62]. Apart from weight gain, antipsychotics may possess markedly negative effects on triglycerides, high-density lipoprotein cholesterol, glucose concentrations, and blood pressure, according to their different pharmacological properties [57]. For instance, the total cholesterol increased with quetiapine, olanzapine, and clozapine, low-density lipoprotein cholesterol increased with quetiapine and olanzapine, triglyceride concentrations increased with quetiapine, olanzapine, zotepine, and clozapine, glucose concentration increased with olanzapine, zotepine, and clozapine, and blood pressure increased with olanzapine and risperidone [57,63].

### 3.2. Neurotransmitters and Hormones of Metabolic Syndrome in Persons with Schizophrenia Treated with Antipsychotics

The theoretical model of antipsychotics-related metabolic disturbances suggests that antipsychotics disrupt metabolic regulation by affecting both the central nervous system that activates hunger, inhibits satiety, and disrupts food rewards as well as impairing the locomotion and metabolism of the peripheral organs [64,65]. Antipsychotics alter glucose and lipid metabolism homeostasis by interfering with the hypothalamus in the central nervous system, as the most important target, and also acting on the liver, pancreatic β-cells, adipose tissue, and skeletal muscle in the periphery [66]. Clinical and preclinical studies have proposed an association between the pharmacological activity of antipsychotics and their functions on metabolic regulation through specific receptor binding, such as the involvement of the serotonin 5-HT2C receptor, dopamine D2 receptor, and the histamine H1 receptor antagonism in increasing food intake and impairing glucose tolerance, serotonin 5-HT2A receptor, serotonin 5-HT6 receptor, muscarinic M3 receptor, and α1 adrenergic receptor antagonism in decreasing insulin and adipocyte lipogenesis, as well as dopamine D2 receptor, dopamine D3 receptor, or serotonin 5-HT1 receptor partial agonism in the disruption of β-cell function and insulin secretion [67,68]. Briefly, in the preliminary proposed mechanisms, antipsychotics hamper the normal functioning of reward-related behaviors and increase food intake through a blockade of the hypothalamic histamine H1 receptor, a blockade of the dopamine D2 receptor, and different polymorphisms of the serotonin 5-HT2C receptor, subsequently affecting the neuropeptides and 5′ AMP-activated protein kinase and producing a supraphysiological sympathetic outflow augmenting the levels of glucagon and hepatic glucose production, resulting in increased blood glucose levels and insulin resistance. The liver then converts the excess exogenous glucose into triglycerides, packages it into low-density lipoprotein cholesterol, and transports it to the adipose tissue. The upregulation of sterol regulatory element-binding protein 2 by antipsychotics could also contribute to imbalance in the lipid metabolism and ultimately result in metabolic syndrome [62,66]. In addition, antipsychotics also directly disrupt the pancreatic islets in insulin secretion from β-cells through the antagonisms of the dopamine D2 receptor and the serotonin 5-HT2C receptor, as well as glucagon secretion from the α-cells through the antagonism of the muscarinic M3 receptor and the serotonin 5-HT2A receptor [62].

Several hormones are also thought to be involved in the management of satiety, feeding, and glucose metabolism, and they have been implicated in the mechanism of antipsychotics-related metabolic disturbances. For example, dysregulations of insulin, cortisol, glucagon, glucagon-like peptide 1, cholecystokinin, adiponectin, ghrelin, leptin, orexin, prolactin, and oxytocin have been observed during treatment with antipsychotics among persons with schizophrenia [69,70]. Although it is still debatable whether this is a compensatory mechanism for insulin resistance or a direct consequence of β-cell stimulation by antipsychotics [66], increased basal insulin secretion at high concentrations was observed in persons with schizophrenia after receiving antipsychotics [71]. Cortisol, the promoter of insulin resistance and hepatic gluconeogenesis, was found to be at higher levels in persons with schizophrenia than in the healthy controls, but it decreased after antipsychotic treatment [72,73]. Antipsychotics apparently also stimulate glucagon secretion in the pancreatic α-cells, resulting in excessive glucose production in the liver [74] even when the peripheral glucose levels are high, and they later develop metabolic syndrome in persons with schizophrenia [66]. Glucagon-like peptide 1, as a stimulator for glucose-dependent insulin secretion that is involved in the regulation of satiety, the inhibition of food intake, and the promotion of body weight loss, was found in higher levels in persons with schizophrenia treated with antipsychotics compared with their healthy controls [75], and it was associated with insulin resistance, obesity, and metabolic syndrome [76]. Cholecystokinin, another neuropeptide that promotes pancreatic secretion and hepatic bile production, is involved in decreasing the rate of gastric emptying for hunger suppression. Cholecystokinin was found to be unchanged after antipsychotic use in persons with schizophrenia [77], although animal studies showed that antipsychotics could counteract the satiating effect of cholecystokinin [78], and the polymorphism of the cholecystokinin receptor gene was found to be associated with antipsychotic-related weight gain [79]. The role of cholecystokinin in the metabolic syndrome of persons with schizophrenia warrants further investigation.

As an adipokine that facilitates lipid and glucose metabolism, adiponectin levels were reported to be lower in the blood concentrations of persons with schizophrenia treated with antipsychotics [80], particularly those suffering from metabolic syndrome [81], compared with their healthy controls. In addition to disruption in reducing food intake, low adiponectin levels in persons with schizophrenia treated with antipsychotics were associated with increased insulin resistance, high blood pressure, hypertriglyceridemia, and low high-density lipoprotein cholesterol [81,82]. Leptin is another adipokine that is involved in appetite and energy balance by inhibiting hunger, increasing energy expenditure, and activating thermogenesis through the production and action of anorexigenic peptides [83]. A recent study in persons with schizophrenia revealed a drop in leptin levels after the initiation of antipsychotics for 6 weeks [84]. However, previous studies noticed an elevation in serum leptin levels among persons with schizophrenia treated with antipsychotics [85], and the evidence suggested that the hyperleptinemic state could be more of a consequence than a cause of antipsychotic-induced weight gain [86]. As it has been proven that leptin levels are positively correlated with the body mass index [87], it is believed that hyperleptinemia that is associated with adipose tissue in persons with obesity without expected appetite reduction may indicate a leptin resistance status [88], a similar concept to insulin resistance in persons with diabetes mellitus. Ghrelin, an orexigenic hormone, increases food intake, modulates glucose homeostasis, decreases brown adipose tissue thermogenesis, and increases reward-seeking behaviors [89], consequently contributing to weight gain and metabolic syndrome. Higher baseline ghrelin levels were associated with higher food cravings [90]. Additionally, the circulating ghrelin levels were negatively correlated with the body mass index [91] and were decreased with weight gain [92]. A recent meta-analysis demonstrated that treatment with antipsychotics decreased the ghrelin levels in persons with schizophrenia [93], concordant with in vitro studies that demonstrated the orexigenic effects of ghrelin [94]. A time-dependent triphasic effect on the serum ghrelin levels by antipsychotics has been discussed. The acute effect of antipsychotics increased the serum ghrelin levels initially in the first week, followed by a secondary downregulation of negative feedback from weight gain within 6 weeks and later returning to previous levels or rising above them during the long-term treatment period [95]. In addition, the presence of two different types of circulating ghrelin, acylated ghrelin and desacylated ghrelin, may also mask the exact relationship of ghrelin with antipsychotics. More precisely, the ratio of acylated ghrelin to desacylated ghrelin was significantly and positively correlated with the body mass index, insulin resistance, and metabolic syndrome among persons with schizophrenia [96].

Orexin, also known as hypocretin, is produced by perifornical neurons and the lateral hypothalamus, and it functions on orexin neurons that are regulated by the availability of glucose, leptin, and ghrelin [97]. Orexin A levels were found to increase in persons with schizophrenia treated with antipsychotics, particularly those treated with less obesogenic antipsychotics [98]. A decrease in orexin A levels was found in persons with schizophrenia treated with olanzapine for 6 weeks [99]. However, the pattern of change in the orexin concentrations of persons with schizophrenia treated with antipsychotics is inconclusive. Contrary to its function in stimulating lipogenesis and increasing food intake by reducing post-ingestion feedback inhibition [97], higher orexin A levels may serve a potential protective role against metabolic syndrome in those persons with schizophrenia by regulating thermogenesis through an increased sympathetic tone and reduced peripheral insulin resistance [100]. Prolactin regulates lipid metabolism through a direct effect on adipose tissue and lipoprotein lipase for the hydrolysis of triglyceride [101]. Furthermore, the indirect effect of prolactin on increasing energy expenditure and decreasing food intake through the dopaminergic pathway also contributes to energy metabolism [102]. Evidence for increased prolactin levels in persons with schizophrenia treated with antipsychotics has been found for decades [103]. Given the role of prolactin in metabolic homeostasis and lipid metabolism, the risk of metabolic syndrome in persons with schizophrenia depends on the prolactin levels that resulted from different antipsychotics [102,104]. Despite advances in pathophysiology and the delineation of risk factors that predispose people to metabolic syndrome by antipsychotics, there are many key aspects that remain unclear.

### 3.3. Neurotransmitters and Hormones of Metabolic Syndrome in Antipsychotic-Naïve Persons with First-Episode Psychosis

In a real-world scenario, not all persons with schizophrenia receiving antipsychotic treatment developed metabolic syndrome. On the contrary, persons with schizophrenia appear to be at significant risk of developing metabolic syndrome regardless of whether they are receiving any antipsychotic medication [105]. A manifestation of metabolic syndrome risk factors in antipsychotic-naïve persons, persons with first-episode psychosis, and unaffected first-degree relatives suggests that metabolic syndrome could be independent of antipsychotic treatments [106,107]. Hence, schizophrenia itself is a risk for the increased onset of metabolic syndrome. The prevalence of metabolic syndrome in antipsychotic-naïve persons with first-episode psychosis is 13.2% worldwide, a 2.52-fold risk for metabolic syndrome over the age- and gender-matched general population [108]. Persons with schizophrenia demonstrate several hallmark features of metabolic syndrome conditions that increase cardiometabolic risk [109,110]. Several studies have reported that drug-naïve persons with first-episode psychosis have significantly higher body mass indexes and abdominal adiposity, increased fasting glucose levels, increased fasting insulin and insulin resistance, and higher total cholesterol and triglyceride concentrations compared with their healthy controls [47,110,111,112,113,114,115]. More specifically, the potential intrinsic link between insulin resistance and impaired glucose tolerance with first-episode psychosis might help to explain the increased prevalence of metabolic syndrome in persons with schizophrenia beyond that with medication, lifestyle, or accessibility of health care [19].

Schizophrenia and metabolic syndrome share several hormonal features that involve glucose regulation and lipid metabolism. As one of the core pathophysiologies for metabolic syndrome, hyperinsulinemia and insulin resistance has also been detected in antipsychotic-naïve persons with first-episode psychosis [116]. Furthermore, lower insulin sensitivity and impaired glucose tolerance were also observed in their unaffected siblings [117,118] compared with the controls. Hyperinsulinemia is not only the consequence of central obesity; it also contributes to the development of obesity [119]. Furthermore, impaired glucose tolerance and insulin resistance among antipsychotic-naïve persons with first-episode psychosis were associated with their symptom severity of schizophrenia [20,120,121]. A hypothesis suggests that the chronic dysregulation of energy metabolism pathways results in neuronal dysfunction, leading to the developmental features reported in schizophrenia, including decreased synaptic plasticity, reduced neuronal size, abnormal glutamate transmission and dopamine release, and myelination deficits [122]. The development of central insulin resistance, which could sit at the intersection of schizophrenia and metabolic syndrome comorbidity, may explain these alterations in dopaminergic reward systems and homeostatic signals affecting food intake, glucose metabolism, and body weight [123].

The hypothalamic–pituitary–adrenal axis provides a further functional link between the central and peripheral control of energy metabolism. Higher circulating cortisol levels and blunted cortisol responses to stress were observed in antipsychotic-naïve persons with first-episode psychosis [122,124,125,126]. High glucocorticoid levels promote gluconeogenesis in the liver and decrease glucose utilization in adipose tissue, resulting in hyperglycemia and insulin resistance. Not only does this have negative consequences on insulin resistance, but high cortisol concentration also has a facilitation effect on elevated triglyceride levels [127]. Leptin, as a hormone that acts as inhibitory feedback for insulin secretion in the adipoinsular axis, suppresses insulin synthesis and secretion by central actions on the hypothalamus and direct effects on the pancreas, and it was found to be dysregulated and in lower levels than in healthy controls among antipsychotic-naïve persons with first-episode psychosis [116,128,129]. A few studies also demonstrated higher leptin levels in antipsychotic-naïve persons with first-episode psychosis [84,130,131,132]. However, hyperleptinemia was also found to be positively correlated with their body mass indexes [132]. Leptin acts as a negative feedback signal by promoting appetite suppression when adipose mass increases to control the energy balance. It is believed that hyperleptinemia that might be induced by weight gain, particularly under antipsychotic treatment [73], contributes to the leptin resistance status, resulting in increased appetites among those subjects [88]. The role of leptin in persons with schizophrenia is the extent of its interaction with dopaminergic regulation. Leptin was found to reduce dopaminergic outbursts in the mesolimbic system [133], and the desensitization of leptin receptors has been associated with the upregulation of dopaminergic genes in the prefrontal cortex [134]. Adiponectin, ghrelin, and orexin have not been consistently demonstrated to have altered levels in antipsychotic-naïve persons with first-episode psychosis [129]. With a very limited sample size, a small number of studies reported no difference in orexin [135] and ghrelin [136,137] in antipsychotic-naïve persons with first-episode psychosis compared with their healthy controls. Previous studies demonstrated a higher level of adiponectin among antipsychotic-naïve persons with first-episode psychosis [84,138], whereas other studies reported no difference compared with healthy controls [128,131,139,140,141]. Higher adiponectin levels may indicate an overly inflammatory status in persons with first-episode psychosis, which contributes to the degenerative process of the brain [138]. Higher levels of prolactin have been found in antipsychotic-naïve persons with first-episode psychosis [136,142,143,144,145]. This finding indicates that hyperprolactinemia in persons with schizophrenia is not solely secondary to dopamine blockades by antipsychotics, but it reflects the potential intrinsic mechanism of the dopamine–prolactin pathway, which might contribute to both schizophrenia and metabolic syndrome [102,146]. In general, subclinical metabolic hormonal anomalies are present at the onset of schizophrenia before their exposure to antipsychotics.

### 3.4. Co-Shared Genetics Pathway in Schizophrenia and Metabolic Syndrome

Given that the genetic pathophysiologies of both schizophrenia and metabolic syndrome have proven challenging to identify, several hypotheses and efforts have been tailored toward disentangling the myth. As of yet, schizophrenia and metabolic syndrome are recognized as multisystem disorders with polygenic architectures [38,147]. The genetic risk of co-sharing of schizophrenia and metabolic syndrome may partly explain the etiology of the crosstalk between these pathways [148]. In recent years, interest in possible schizophrenia–metabolic syndrome risk co-sharing and the evidence of overlapping gene loci has increased. Several genome-wide association analysis studies detected the shared loci of schizophrenia and metabolic disturbance [149,150]. For example, the polygenic risk score related to the onset of schizophrenia is associated with insulin resistance in antipsychotic-naïve persons with first-episode psychosis [113,151]. The main findings of the genetic commonality between schizophrenia and metabolic syndrome are derived from studies on persons with type 2 diabetes mellitus, with numerous shared candidate genes that are predisposed to both disorders [152,153]. Previous family-based genome-wide linkage studies identified a number of overlapping risk loci between schizophrenia and diabetes, including chromosomes 1p13, 1p36, 1q21–24, 1q25, 2p13–22, 2q14, 2q33, 2q36, 3p22, 3q29, 4q25, 5q13, 6p21, 6q25, 7p15, 7p21, 7q21, 7q31, and 9p24 [154,155], which are the gene-rich regions that harbor multiple common candidate genes for susceptibility to these disorders [156]. Possible pleiotropic genes identified for targeting the association with metabolic syndrome susceptibility in persons with schizophrenia include the catechol-O-methyltransferase gene (*COMT*), endocannabinoid receptor type 1 gene (*CNR1*), brain-derived neurotrophic factor gene (*BDNF*), methylenetetrahydrofolate reductase gene (*MTHFR*), serotonin receptor 2A gene (*HTR2A*), serotonin receptor 2C gene (*HTR2C*), fat mass and obesity-associated gene (*FTO*), leptin gene (*LEP*), leptin receptor gene (*LEPR*) [157], cyclin-dependent kinase 5 regulatory subunit-associated protein 1-like 1 gene (*CDKAL1*) [158], nitric oxide synthase 3 gene (*NOS3*) [159], solute carrier family 2 member 2 gene (*SLC2A2*), forkhead box O3 gene (*FOXO3*), and furin gene (*FURIN*) [160].

More specifically aiming to detect the gene risk variants co-shared by schizophrenia and specific factors of metabolic syndrome, several gene loci were examined in previous studies. A majority of the studied gene loci are focused on an association with glucose metabolism and insulin resistance, including the dopamine 2 receptor gene (*DRD2*) [161], insulin-like growth-factor-2 mRNA-binding protein-2 gene (*IGF2BP2*), transcription factor 7 like 2 gene (*TCF7L2*) [162], disrupted-in-schizophrenia 1 gene (*DISC1*) [163,164], transmembrane protein 108 (*Tmem108*) [165], ADP ribosylation factor like GTPase 6 interacting protein 4 gene (*ARL6IP4*), 2-oxoglutarate and iron dependent oxygenase domain-containing 2 gene (*OGFOD2*), phosphatidylinositol transfer protein membrane-associated 2 gene (*PITPNM2*), cyclin-dependent kinase 2-associated protein 1 gene (*CDK2AP1*), ATP-binding cassette subfamily B member 9 gene (*ABCB9*), M-phase phosphoprotein 9 gene (*MPHOSPH9*), sterol regulatory element-binding transcription factor 1 gene (*SREBF1*), target of Myb1-like 2 membrane-trafficking protein gene (*TOM1L2*) [166], and phospholipase A2 group IVA (*PLA2G4A*) [167]. Only a few gene loci are discussed with regard to other metabolic parameters, including the VRK serine/threonine-protein kinase gene (*VRK2*), 5’-nucleotidase cytosolic II gene (*NT5C2*), INO80 complex subunit E gene (*INO80E*), yippee-like 3 gene (*YPEL3*), mitogen-activated protein kinase 3 gene (*MAPK3*) [166], β3 adrenoceptor gene (*ADRB3*) [168], neurexin 3-alpha gene (*NRXN3*) [169], insulin-induced gene 2 gene (*INSIG2*), and melanocortin 4 receptor gene (*MC4R*) [170] for the body mass index and central obesity, as well as the solute carrier family 39 member 8 gene (*SLC39A8*), autophagy and beclin 1 regulator 1 gene (*AMBRA1*), lysine methyltransferase 5A gene (*KMT5A*), GATA zinc finger domain-containing 2A gene (*GATAD2A*), transmembrane 6 superfamily member 2 gene (*TM6SF2*) [166], activator of transcription and developmental regulator gene (*AUST2*) [171], and apolipoprotein A1 gene (*ApoA1*) [172] for high-density lipoprotein cholesterol and triglyceride metabolism. Among the investigated gene loci, the genes involved in the dopamine–prolactin pathway, including the neuropeptide Y gene (*NYP*) [173] and prolactin gene (*PRL*) [174], may also attract attention to their contribution to the comorbidity of schizophrenia and metabolic syndrome [102,148,156]. Together, the above-mentioned findings suggest a molecular commonality between schizophrenia and metabolic syndrome that may influence each other and that could be partly attributable to shared genes [160]. Some researchers have argued that since a dysfunction pathway in schizophrenia may lead to metabolic syndrome over time regardless of the use of antipsychotics, the causality hypothesis may more strongly impact the comorbidity of schizophrenia than the pleiotropic gene theory [154,175]. The need to investigate the depth of the genetic neuro-psycho-metabolic pathways is warranted.

## 4. Oxytocin

### 4.1. Love Hormone: Oxytocin

A growing effort has been put forth to establish a biological foundation in the connections between schizophrenia and metabolic syndrome. Though conflicting results have been reported [176], several biomarkers that appeared to be associated with these two disorders at the level of central or peripheral biomarkers make it likely that metabolic syndrome and schizophrenia are connected via common pathways that are central to schizophrenia pathogenesis. One of the investigated biomarkers is oxytocin. Oxytocin, known as a love hormone that promotes love, lust, and labor, has been best known for its roles in childbirth and breastfeeding. The oxytocin gene (*OXT*) encodes a precursor protein that is processed to produce oxytocin and its protein carrier neurophysin I. The inactive precursor oxytocin protein is carried by neurophysin I and is hydrolyzed and released as active oxytocin after being catalyzed by peptidylglycine alpha-amidating monooxygenase [177]. In human physiology, the expression signals arise from nipple suckling and cervical stretching and reach the magnocellular neurosecretory cells of the paraventricular nuclei and supraoptic nuclei through the dopaminergic afferent pathways that originate in the arcuate and periventricular nuclei. Apart from the dopaminergic signals, oxytocinergic neurons in the magnocellular neurosecretory cells also receive noradrenalinergic afferent projections from the A1 and A2 cell groups of the medulla oblongata, serotonergic afferent projections originating from serotonin neuronal cell bodies in mesencephalic dorsal raphe nuclei, and GABAergic pathways from the limbic system [178]. Trafficking of the neurokinin 3 receptor in oxytocinergic neurons occurs in the response to the regulation of genes involved in the increased secretion of oxytocin during lactation and parturition [178]. The nine-amino acid hydrophilic cyclic hormonal neuropeptide oxytocin is produced by the magnocellular neurosecretory cells of the paraventricular nuclei and supraoptic nuclei of the hypothalamus, which are transported and stored in Herring bodies at the axon terminal and released by neurohypophysis of the posterior pituitary into circulation [179]. The release of oxytocin into circulation is in response to action potentials that depolarize the axon terminal by the opening of voltage-dependent calcium channels inducing exocytosis of the secretory granules [178]. In addition to the peripheral pathway, paraventricular nuclei have specific projections, particularly through axons that collaterally innervate neurons in the amygdala, striatum, substantia nigra, hypothalamus, hippocampus, and nucleus accumbens [179]. Evidence of an increase in the measured oxytocin in the cerebrospinal fluid after peripheral administration proves the potential of oxytocin to act centrally [180,181]. Oxytocin is transported by the receptor for advanced glycation end products in brain capillary endothelial cells to cross the blood–brain barrier [182]. In humans, oxytocin-producing neurons are also expressed in the myenteric and submucous ganglia, intestines, bone marrow osteoblasts, liver, and subcutaneous adipose tissue [183]. Pulsatile oxytocin secretion has been demonstrated in animal studies, but the secretion patterns in humans are still debatable. However, recent studies have failed to find the time-of-day differences or diurnal rhythms of oxytocin in human cerebrospinal fluid, blood, or saliva [184,185]. Oxytocin is degraded by the M1 family of aminopeptidases that are expressed in different cell types, including adipocytes and leukocytes [186].

The oxytocin receptor is a seven-transmembrane domain metabotropic G-protein-coupled receptor that is encoded by the oxytocin receptor gene (*OXTR*). There are two different types of oxytocin receptors that have been identified, namely type Gq and type Gi, according to their actions either through protein-coupled Gq or Gi, respectively. Oxytocin receptors are found in multiple human brain regions, including the basal ganglia, hypothalamus, septal nuclei, amygdala, hippocampus, the nucleus of the solitary tract, spinal cord, and multiple cortical regions, as well as other tissues, including the uterus, breast, aorta, esophagus, the nodose ganglion of the vagus nerve, the myenteric plexus of the gastrointestinal tract, and adipocytes [183]. Oxytocin receptor type Gq is expressed mostly in the central nervous system, whereas type Gi is abundantly expressed in the peripheral organs and tissues but is restricted in the central nervous system [187]. The activation of oxytocin receptor type Gq/11 by oxytocin contributes to the phosphorylation of proteins through the regulation of ligand-gated ionotropic receptors, ligand-induced trafficking of the G-protein-coupled receptors to the nucleus, and different ion channels [178]. On the other hand, the activation of oxytocin receptor type Gi induces an inhibition of adenylate cyclase activity and decreases the concentration of cAMP, which later contributes to the regulation of membrane excitability, the synthesis and release of neurotransmitters, and synaptic plasticity [187].

Oxytocin and its receptor are known for orchestrating a broad repertoire of physiological processes across species. It acts as a neurotransmitter in the brain’s regulation of a range of physiological processes, including parturition, lactation, cell growth, and wound healing [42]. Recent studies have demonstrated the influence of oxytocin on a range of human behaviors, including the recognition of facial emotions, empathy, confidence, parent–child interactions, attachment, bonding, stress, sexual response, cognition, and eating behavior [188]. Exogenous oxytocin was proven to enhance economic cooperation and trust, increase the sense of belonging to a group, lower competitiveness toward in-group members, and increase the salience of available external social cues that are all considered prosocial behaviors in a context-dependent manner [189]. The involvement of the oxytocinergic system in the formation of social cognition, working memory, spatial memory, and episodic memory, which are mediated by the brain networks of the hippocampus, amygdala, and prefrontal cortex, has been reported [190]. The regulation of cognitive functions is thought to be associated with the glutamatergic system that is modulated by oxytocin, as oxytocin binds to the presynaptic membrane of glutamatergic neurons and promotes glutamate release [191]. Meanwhile, a reduction in GABA release by oxytocin through the activation of presynaptic oxytocin receptors was also observed in the same study [191]. The involvement of oxytocin in the integration of social and spatial memory in the hippocampus and in the execution control of working memory has been reported in various animal studies [190]. The majority of human studies are focused on facial emotional recognition, which is a subcomponent of social cognition, although the results are inconsistent [192]. A human study that examined the enhancement of oxytocin on the placebo effect demonstrated that intranasal oxytocin induces the expectancy-driven enhancement of working memory [193]. Furthermore, the role of oxytocin in neuroprotection has been discussed. Oxytocin release is stimulated by psychological stress, whereas exogenous oxytocin administration significantly reduced the glucocorticoids that respond to stress (i.e., cortisol) [194] and reduced the molecular responses of the hypothalamus–pituitary–adrenal axis, acting as hormonal protection from proinflammatory cytokines induced by glucocorticoids [178].

Various confounding factors that influence oxytocin modulation should be discussed. Oxytocin is a gender-specific hormone. The menstrual cycle and the use of contraceptives are associated with the concentrations of circulating oxytocin. Oxytocin is lower in the early to mid-follicular phase of the menstrual cycle compared with the levels measured at other menstrual cycle phases [195]. Since the post-menopausal years are associated with an increased risk of metabolic syndrome independent of age [196], and as oxytocin affects reproductive-age women differently than it does postmenopausal women [197], this evidence suggests that oxytocin central signaling may also be modulated by estrogen [198]. In addition, personal characteristics and environmental factors, including eating attitudes and dietary habits [199], aging [200], chronic diseases [201], traumatic events [202], and childhood trauma [203], have also been proven to have certain influences on oxytocin modulation. Aside from that, oxytocin stimulates prolactin secretion, initiating an oxytocin–prolactin positive feedback loop [204], and furthermore, oxytocin modulates the dopamine–prolactin pathway and may confer risk for prolactin pathway disruption and metabolic syndrome comorbidity [148]. The interactions of oxytocin and dopamine were illustrated in the animal studies of intranasal oxytocin administration that reduced the dopamine-associated conditioned food reward-directed behaviors [205], as well as oxytocin administration that increased dopamine release in the nucleus accumbens in response to enhanced prosocial behaviors [206]. The interactions of oxytocin with other hormones, including testosterone, insulin, irisin, ghrelin, and arginine vasopressin [207,208,209,210] should also be considered. It has been reported that the systemic administration of cholecystokinin octapeptide induced a dose-dependent increase in oxytocin levels [178]. The interaction of antipsychotics with oxytocin, particularly in persons with schizophrenia, is also widely discussed. Apart from interactions with other neurotransmitter systems, including dopamine, serotonin, and GABA, several antipsychotics are known to interact with the oxytocin system. For example, amperozide and clozapine increase the plasma levels of oxytocin, whereas quetiapine and clozapine also increase cFos activation in oxytocin cells in the paraventricular nuclei [211,212]. Other psychoactive agents may also interfere with the metabolically relevant signals of oxytocinergic neurons, including caffeine [213] and nicotine [214].

Owing to the diversity of oxytocin and the expression of its receptor, the dysregulation of oxytocin is proposed as a candidate for pathophysiology and novel intervention in the treatment of conditions as apparently diverse as autism spectrum disorders, schizophrenia, postpartum depression, anxiety, post-traumatic stress disorders, addiction, pain, metabolic disorders, diabetes, cardiovascular diseases, cancer, and infectious diseases [215]. The growing interest in studying the implication of therapeutic use further accelerates the understanding of the role of oxytocin in the pathophysiology of those disorders, particularly regarding specified symptoms. For instance, exogenous oxytocin did not improve general symptomatology, but it improved repetitive behaviors and social function in persons with autism spectrum disorder [189]. Although it is not possible to completely understand the role of oxytocin, given that the findings of individual studies were complex and often contradictory, oxytocin might still play an important role in the aforementioned disorders, particularly disorders that are characterized by deficits in social functioning (i.e., schizophrenia) [216].

### 4.2. The Implication of Oxytocin in Schizophrenia

Oxytocin as a hormonal treatment for persons with schizophrenia has been gaining increasing attention over the past few decades, particularly regarding its association with the pathophysiology, symptomatology, and treatment of schizophrenia. Previous studies have found oxytocinergic dysfunction in persons with schizophrenia [217,218]. A hypothetical neurofunctional model posits that abnormal dopaminergic and oxytocinergic reward system signaling in the amygdala engenders a neural milieu that improperly assigns emotional salience processing and leads to misguided social responses, ranging from withdrawal and isolation to suspicion and paranoia in persons with schizophrenia [179]. Neuroimaging studies revealed that central oxytocinergic activities in several brain regions are responsible for the pathophysiology of schizophrenia. Oxytocin-sensitive basal ganglion networks that overlap with dopaminergic modulation areas provide indirect evidence for interactions between oxytocinergic and dopaminergic systems that are responsible for the regulation of salience processing, approaches, and motivation [219]. Oxytocin attenuates the brain activity of the amygdala, fusiform gyrus, and anterior cingulate gyrus in response to negative emotional faces [220]. Furthermore, the peripheral basal oxytocin levels were found to be associated with the oxytocinergic activities of the middle and superior frontal cortex, cingulate cortex, cerebellum, and thalamus in persons with schizophrenia [221]. The administration of exogenous oxytocin was found to attenuate normal bias in mentalizing and the processing of facial emotion, salience, aversion, uncertainty, and ambiguity in the social stimuli of persons with schizophrenia, particularly that which contributes to negative symptoms, through reduced activation in the brain networks of the amygdala, temporoparietal junction, posterior cingulate cortex, precuneus, and insula [222,223].

On a genetic basis, the involvement of single-nucleotide polymorphisms of the oxytocin gene (*OXT*) in schizophrenia vulnerability has been proven [224]. In addition, single-nucleotide polymorphisms of the oxytocin receptor gene (*OXTR*) are also associated with a risk of schizophrenia [225], and a relationship with clinical symptomatology in schizophrenia, including general psychopathology [226], negative symptoms [224], social cognition [227,228], and the treatment response with antipsychotics for positive symptoms [226], has been more clearly elucidated. The single-nucleotide polymorphisms of the oxytocin receptor gene could profoundly affect treatment-refractory symptoms by mediating social cognitive processes in persons with schizophrenia [217].

On the hormonal level, a lower peripheral oxytocin level was found in persons with first-episode psychosis [229,230] and in persons with schizophrenia [188,218,231,232], regardless of whether antipsychotics were administered or not, compared with their healthy controls. The association of endogenous oxytocin levels with the symptoms of persons with schizophrenia has been proven. For instance, lower oxytocin levels in persons with first-episode psychosis are associated with a poor premorbid IQ and worse sustained attention [229]. The levels of oxytocin are also negatively correlated with the severity of positive symptoms, negative symptoms, and social cognition in persons with schizophrenia [8,233], as well as predicting a worse outcome with a lower oxytocin level [188,234]. Recent studies demonstrating the role of oxytocin in the pathophysiology of schizophrenia and its tolerability when delivered intranasally have sparked interest in its therapeutic potential. With the utmost priority, the intranasal oxytocin produces no reliable side effects and is not associated with adverse outcomes when delivered for short-term use in controlled research settings [235]. The benefits of exogenous oxytocin on the positive symptoms, negative symptoms, and cognitive function of persons with schizophrenia have been demonstrated [189], particularly with a higher dose of adjunctive intranasal oxytocin [236]. The benefits of intranasal oxytocin have been extended to the social cognition domains of persons with schizophrenia [8] not only in emotion recognition [237] but also in higher-order social cognition processes that include theory of the mind [238] and empathy [239]. By contrast, not all studies detected significant differences in the levels of oxytocin between individuals both with and without schizophrenia [240,241], and several studies failed to reveal the benefits of oxytocin treatment in persons with schizophrenia [242,243,244]. The optimal administration scheme of intranasal oxytocin that involves repeated applications is intensely debated [245]. Long-term exposure to high concentrations of oxytocin is hypothesized to downregulate the oxytocin-signaling machinery. It attenuates the oxytocinergic effect on brain function, and smaller improvements in the general symptoms of persons with schizophrenia have been reported [246], as demonstrated by the dampening of the response of the amygdala negatively affecting healthy volunteers who received daily intranasal oxytocin for 1 week compared with those who received single doses [247]. The mechanisms underpinning both schizophrenia and the oxytocinergic system have yet to be adequately characterized. The potential therapeutic effects of oxytocin, similar to antipsychotics, are exerted at the mesolimbic level through the inhibition of dopaminergic mechanisms [248]. Undoubtedly, the interactions of oxytocin with other functional systems are complex, and further work is needed to understand the implications and mediations of oxytocin in schizophrenia [8].

### 4.3. Metabolic Effects of Oxytocin

There is growing evidence that oxytocin has important metabolic effects. The central and peripheral targets of brain oxytocin systems in metabolic regulation are illustrated in Figure 1. Oxytocin-producing neurons of the paraventricular nuclei and supraoptic nuclei act as nutrient status sensors, and they are sensitive to nutrients and hormones, including cholecystokinin, glucagon-like peptide 1, leptin, and insulin [183]. By increasing the sensitivity of hindbrain nucleus tractus solitarius neurons to satiety signals, leptin, as an example of a nutrient status signal, activates the oxytocin-producing parvocellular neurons of the paraventricular nuclei [249] and magnocellular neurons that have axonal projections to the arcuate nucleus [250] and subsequently induces satiety and decreases food intake. Furthermore, oxytocin signaling to the nucleus tractus solitarius neurons also increased the expenditure and net energy balance by heightening sympathetic nervous system activity simultaneously [251]. To summarize, oxytocin has multiple roles that are exerted both centrally and peripherally through secretion from the neurohypophysis and by local paracrine actions at the gut in energy balance and metabolism. Oxytocin inhibits food intake through the actions of the satiety-regulating centers of the hypothalamus, at the sweet taste receptors, and in the rewards system of the brain. Oxytocin also increases energy expenditure and lipolysis, slows gastric motility, and improves glucose tolerance by its actions in the pancreas [252,253].

Oxytocin-knockout and oxytocin receptor-knockout mice demonstrated decreased insulin sensitivity, increased glucose excursions, increased abdominal fat pads, and increased triglycerides [254,255]. The mechanism of oxytocin in decreasing appetite and food intake is conveyed through a modulating effect on reward-driven eating by the expression of oxytocin receptors in the ventral tegmental area, as well as in nucleus accumbens [256] and from projections from oxytocin-producing neurons in the paraventricular nuclei and supraoptic nuclei in the hedonic pathways, in addition to affecting the positive-valence mechanism at the central amygdala [257]. Impairments in oxytocinergic signaling are associated with increased carbohydrate- and fat-rich meals in animal studies, implicating a physiological role for oxytocin in food intake [245]. The administration of oxytocin in animal studies has demonstrated promising positive results in metabolic parameters, including reductions in caloric intake, weight, adipocyte size, and body fat, as well as the facilitation of fat oxidation and lipolysis, the reuptake of triglyceride, and altered satiety [245]. That aside, exogenous oxytocin also decreases food intake, increases energy expenditure, and improves glucose tolerance and insulin resistance, in addition to inducing weight loss in animal models of diet-induced obesity and diabetes [245,258]. More specifically, a reduction in food intake was found when oxytocin was administrated directly into the arcuate nucleus, ventromedial hypothalamic nucleus, ventral tegmental area, and nucleus tractus solitarius, which are full of oxytocin receptors and which receive oxytocinergic neurons projections from the paraventricular nuclei and the supraoptic nuclei [259].

In addition to affecting energy intake, oxytocin also modulates energy expenditure and body composition. For example, animal studies have proven that oxytocin treatment increased energy expenditure in addition to producing more weight loss [258], fat mass loss, and visceral fat loss [260] when food intake was not altered. More promisingly, oxytocin-mediated weight loss was maintained after the effects of oxytocin waned [261]. In lipid metabolism, oxytocin reduces triglycerides and increases free fatty acids in cultured adipocytes [262], suggesting that oxytocin may reduce body weight by increasing lipolysis through a direct effect on adipocytes or an indirect effect on white adipose tissue via polysynaptic sympathetic nervous system circuits [245]. Oxytocin promotes white adipose tissue lipolysis and brown adipose tissue thermogenesis by increasing sympathetic nervous system activity [260,262]. Oxytocin has also been proven to regulate the sympathetic nervous system in the stimulation of energy expenditure through polysynaptic projections from oxytocinergic neurons in the paraventricular nuclei to interscapular brown adipose tissue and the stellate ganglia [263]. The injection of oxytocin in the median raphe and the ventromedial hypothalamic nucleus increased the heart rate and body temperature, in addition to stimulating energy expenditure in animal studies [264]. The peripheral effects of oxytocin signaling are also involved in energy metabolism through the suppression of gastric emptying, including its action as a brake on intestinal motility, decreasing the mucosal activation of enteric neurons, promoting enteric neuronal development, regulating the proliferation of crypt cells and mucosal permeability, and protecting against inflammation [265]. Oxytocin also suppresses gastric emptying through the release of cholecystokinin octapeptide [266] and the suppression of ghrelin [210] indirectly. Aside from that, an cardioprotection effect from oxytocin by reducing the progression of atherosclerosis has been reported [267]. Oxytocin modulates arterial pressure by acting on the nucleus of the solitary tract that contains vagus nerve synapses with a high concentration of oxytocin receptors. In an animal study, the stimulation of oxytocinergic neurons improved the autonomic control of blood pressure [268]. Oxytocin affects the heart rate, blood vessels, and kidneys directly, or it reduces the sympathetic activation of vasoconstriction and promotes vasodilation with parasympathetic nerves indirectly through the atrial natriuretic peptide receptor, nitric oxide receptor, and α2-adrenergic receptor [178,269].

The results from both endogenous oxytocin secretion and exogenous oxytocin treatment in human studies support the hypothesis of preclinical data, indicating that oxytocin is involved in food intake and the metabolic homeostatic pathway, but a clear consensus has not yet emerged. A dysfunction of the oxytocinergic system has proven its role in metabolic mechanisms in human genetic studies. Single-nucleotide polymorphisms of the oxytocin receptor gene, including rs237878, rs237885, rs2268493, rs2268494, rs2254298, rs53576, and rs2268498, have been proven to be associated with overeating, sweet and fatty food preferences, and reward and punishment sensitivity, which are linked to glucose metabolism and energy metabolism [270]. Evidence of the endogenous oxytocinergic dysfunction that leads to impaired satiety and obesity is well-established in persons with Prader–Willi syndrome [271,272], who had a reduced number and size of oxytocin-producing parvocellular neurons that are associated with a reduction in oxytocin secretion [273]. The metabolic effects of oxytocin have further been elucidated in persons with obesity and metabolic syndrome. Lower oxytocin levels in adults with obesity [274,275] and food addiction [276] were found compared with their controls. In addition, decreased levels of oxytocin have also been reported in adults with type 2 diabetes mellitus and children with metabolic syndrome [209,277]. A significant positive correlation was also found between circulating oxytocin and the indices of metabolic syndrome, including fasting glucose, insulin levels, impaired glucose tolerance, homeostasis model assessment-estimated insulin resistance, triglycerides, low-density lipoprotein cholesterol, total cholesterol, and central obesity [278,279]. However, in other studies, higher peripheral oxytocin concentrations were positively associated with lower fasting glucose, lower insulin levels, higher body mass indexes, and visceral fat [277], and the positive correlation remained significant after controlling for age [280].

The implications of oxytocin in metabolic regulation are extended to exogenous oxytocin application in human studies. Contrary to several negative results [281,282], intranasal oxytocin treatment reduced the consumption of sweet foods [283,284], total caloric intake [285], reward-driven food intake [286], and food cravings [287] in humans, particularly in persons with obesity [288,289]. At the hormonal level, exogenous oxytocin treatment attenuated the peak excursion of plasma glucose and increased insulin secretion in response to the intravenous glucose tolerance test in healthy men [290]. Oxytocin also demonstrated a favorable effect on human metabolic parameters. For example, intranasal oxytocin treatment showed a significant reduction in body weight loss, total cholesterol, and low-density lipoprotein in obese but nondiabetic adults [291]. The beneficial effect of exogenous oxytocin was also observed in persons with leptin-resistant obesity [292]. A recent meta-analysis demonstrated the efficacy of intranasal oxytocin in reducing food intake in persons without psychiatric disorders [293]. Aside from that, intranasal oxytocin was found to result in an increase in cholecystokinin octapeptide preceding the suppression of food intake [285]. An improvement in insulin sensitivity and insulin resistance and an increase in fatty acid oxidation after intranasal oxytocin use were also reported in that same study [285]. Other than decreased food consumption, intranasal oxytocin also attenuates the basal and postprandial levels of adrenocorticotropic hormone and cortisol in conditions that curb a meal-related increase in plasma glucose, proving the role of oxytocin in the hypothalamic–pituitary–adrenal axis activity and the glucoregulatory response to food intake in humans [286,288]. In a study of functional magnetic resonance imaging for the food motivation paradigm, intranasal oxytocin blunted the ventral tegmental area activation of the participants toward high-calorie food stimuli as a reference to non-food visual stimuli [294]. Following oxytocin administration, hypoactivation of the brain regions that are involved in additional hedonic (ventral tegmental area, orbitofrontal cortex, insula, globus pallidus, hippocampus, and amygdala) [294,295] and homeostatic food motivation (hypothalamus) [296] were observed in neuroimages of humans. The activation of the dopaminergic reward-processing circuits that are involved in generalized rewards (caudate nucleus and putamen) [297] and cognitive control (middle and superior frontal gyrus, precuneus, anterior cingulate gyrus, prefrontal cortex, and supplementary motor area) was observed [287,294]. The aforementioned evidence, although at a preliminary stage, provides a potential neurobiological mechanism for anorexigenic oxytocin effects in humans.

## 5. Oxytocin Dysfunction as a Common Mechanism Underlying Schizophrenia and Metabolic Syndrome

The heightened risk of metabolic syndrome is independent of antipsychotics potentially impairing metabolism among persons with schizophrenia, as the metabolic syndrome risk factors are obviously detected in antipsychotic-naïve persons, persons with first-episode psychosis, and their unaffected first-degree relatives. Early evidence supports the role of oxytocin system dysfunction in both schizophrenia and metabolic syndrome. Overlapping neurobiological profiles of metabolic risk factors and psychiatric symptoms suggest that oxytocin system dysfunction may be one common mechanism underlying schizophrenia and metabolic syndrome [106]. Schizophrenia and metabolic syndrome share a similar pattern of oxytocinergic dysfunction. Lower basal oxytocin levels among persons with schizophrenia are associated with a higher risk for metabolic syndrome, and the indices of metabolic syndrome are significantly correlated with circulating oxytocin levels. In concordance, persons with metabolic syndrome also demonstrated lower basal oxytocin levels that were highly correlated with their metabolic parameters. Exogenous oxytocin treatment is effective in both populations, although at a preliminary status. For example, intranasal oxytocin improved social cognition in persons with schizophrenia and reduced food intake in persons with metabolic syndrome. In addition, in a double-blind crossover study examining satiety signaling, a decrease in leptin was found after intranasal oxytocin administration in persons with schizophrenia [298]. The authors suggested that the modulation of leptin by oxytocin may have effects on not only the metabolic parameters but on the treatment of schizophrenia.

Persons with metabolic syndrome are associated with the presence of cognitive dysfunction across their lifespans [299], while neurocognitive deficits are one of the hallmark symptoms of those with schizophrenia [300]. For instance, metabolic syndrome is significantly associated with cognitive impairment in persons with schizophrenia [301]. Persons with metabolic syndrome perform considerably worse at cognitive performance tasks compared with those with no metabolic syndrome [302]. Metabolic syndrome was associated with a higher risk for cognitive impairment that was observed across several studies. More particularly, higher triglyceride levels, higher systolic blood pressure, and a greater waist circumference were the strongest predictors of cognitive function in persons with schizophrenia [303], comparable with those of persons with metabolic syndrome [304]. The role of oxytocin in cognitive impairment through its regulation on the hippocampus, amygdala, and prefrontal cortex has been demonstrated in studies on oxytocin-knockout mice and studies on oxytocin receptor polymorphism in humans. An improvement in working memory after intranasal oxytocin treatment has also been proven in persons with schizophrenia [305].

Preceding autonomic nervous system dysfunction in persons with metabolic syndrome [306] as well as in those with schizophrenia [307] is another neurobiological factor that is common to schizophrenia and metabolic syndrome. Autonomic nervous system dysfunction has also been proven to be associated with or predict insulin resistance and obesity [308,309]. Oxytocin regulates both the sympathetic and parasympathetic nervous systems by the action of oxytocin receptors in the vagus nerve synapse of the nucleus of the solitary tract, as well as through indirect action via other neuropeptides (i.e., atrial natriuretic peptides). Furthermore, oxytocin receptor gene polymorphism and exogenous administration have also been shown to have a significant effect on autonomic nervous system regulation [310]. Interestingly, higher levels of loneliness were associated with diminished parasympathetic cardiac reactivity in response to intranasal oxytocin [311]. Autonomic nervous system dysfunction could inhibit the ability to appropriately approach or withdraw in social situations [312], which has obviously been observed among persons with schizophrenia that suffer from social dysfunction. Therefore, another possible clue that raises the speculation on oxytocin in linking schizophrenia and metabolic syndrome lies in loneliness and social isolation. Apart from social stigma and discrimination, high rates of loneliness and social isolation [313] among persons with schizophrenia could be explained by their social cognitive dysfunction [314], which may be partly modulated by oxytocin. Increased loneliness was found to be associated with an increased risk of metabolic syndrome in persons with psychosis [315]. Additionally, it is plausible that the social effects of oxytocin could be related to its metabolic effects, such as by influencing eating patterns, responses to stress, and physical activity, all of which are complex behaviors with social dimensions [183].

A wealth of research has found desynchrony in the resting state default mode network of persons with metabolic syndrome [316] and also those with schizophrenia [317]. Although the actual dynamic effect is unclear, oxytocin and its related genes are associated with the maturity of the default mode network [318,319]. The authors proposed that a greater inborn functionality of the oxytocinergic system sustains the maturation of the default mode network and buffers the effects of chronic stress on default mode network connectivity [318]. Chronic stress, particularly childhood trauma, has detrimental consequences that lead to metabolic syndrome [320] and schizophrenia [321], which has been proven in the genetic changes of persons with metabolic syndrome [322] and persons with schizophrenia [203]. These interactions are also illustrated through the association between childhood trauma and metabolic syndrome in persons with schizophrenia [137,323]. Recent findings postulate that schizophrenia risk loci can influence stochastic variation in gene expression through epigenetic processes, highlighting the intricate interaction between the genetic and epigenetic control of neurodevelopmental trajectories, leaving molecular scars that influence brain functions [324]. This is in concordance with the recent allostatic theory of oxytocin, which indicates that the oxytocinergic system acts as an important role in maintaining stability in changing environments at each developmental stage to promote survival [325]. In connection with oxytocin, the balance in energy regulation and the ability to adapt to the social environment are all crucial in survival. Dysfunction of the oxytocinergic system can not only lead to brain dysfunction in persons with schizophrenia but also hormonal dysfunction in persons with metabolic syndrome.

In recent years, interest in possible schizophrenia and metabolic syndrome risk co-sharing, and the evidence of overlapping loci has increased. Several genes have shown strong evidence for association, including the genes that are involved in the oxytocinergic system. An analysis of random gene sets of half a million participants suggests that the oxytocin system has pleiotropic effects on both social and metabolic phenotypes by providing evidence for the involvement of the oxytocin-signaling pathway in the shared genetic liability of schizophrenia and metabolic syndrome [326]. In the pairwise correlation and clustering analysis, the food intake cluster (energy intake, sugar intake, and food weight) was positively associated with oxytocin-specific polygenic scores, and the body fat cluster (body mass index, trunk fat, and total body fat) pointed to a positive association with the polygenic scores of oxytocin, diabetes, and schizophrenia [326]. Additionally, although not fully proven in schizophrenia, oxytocin receptor gene polymorphism, including rs53576 [327] and rs2254298 [328], was found to play an important role in glucose homeostasis [326].

## 6. Conclusions

Increasing evidence for connections between the psychopathology of schizophrenia and the regulation of metabolism makes it likely that metabolic syndrome and schizophrenia are connected via common pathways that are central to schizophrenia pathogenesis, which may be underpinned by oxytocin system dysfunction. Oxytocin is an anorexigenic hormone that involves homeostatic satiety-related signaling, metabolic modulation, and a reduction in feeding rewards. Oxytocin system dysfunction may partly explain this piece of the puzzle for the mechanism underlying this association. However, there are still abundant pressing questions surrounding the etiology and manifestation of oxytocin system dysfunction in persons with schizophrenia with respect to metabolic syndrome. Are schizophrenia and metabolic syndrome two distinct disorders, or is schizophrenia a multisystem disorder? Compared with their healthy controls, persons with first-episode psychosis experienced significant alterations in their immune parameters, cardiometabolic parameters, hypothalamic–pituitary–adrenal axis parameters, brain structure, neurophysiology, and neurochemistry [329]. This suggests that early in psychosis, dysfunction is present across multiple organ systems among persons with first-episode psychosis. On this account, schizophrenia may be a multisystem disorder where one organ system is predominantly affected and where other organ systems are also concurrently involved. With regard to the multisystem disorder hypothesis, schizophrenia and metabolic syndrome could share common dysfunctional pathways. However, the mechanisms underpinning the shared risk for schizophrenia and metabolic syndrome remain unclear. Altogether, the shared effects of oxytocin system dysfunction help with providing a better understanding of the co-occurrence of schizophrenia and metabolic syndrome and with proposing a new mechanistic insight into schizophrenia pathogenesis and treatment.

## Figures and Tables

**Figure 1 ijms-23-07092-f001:**
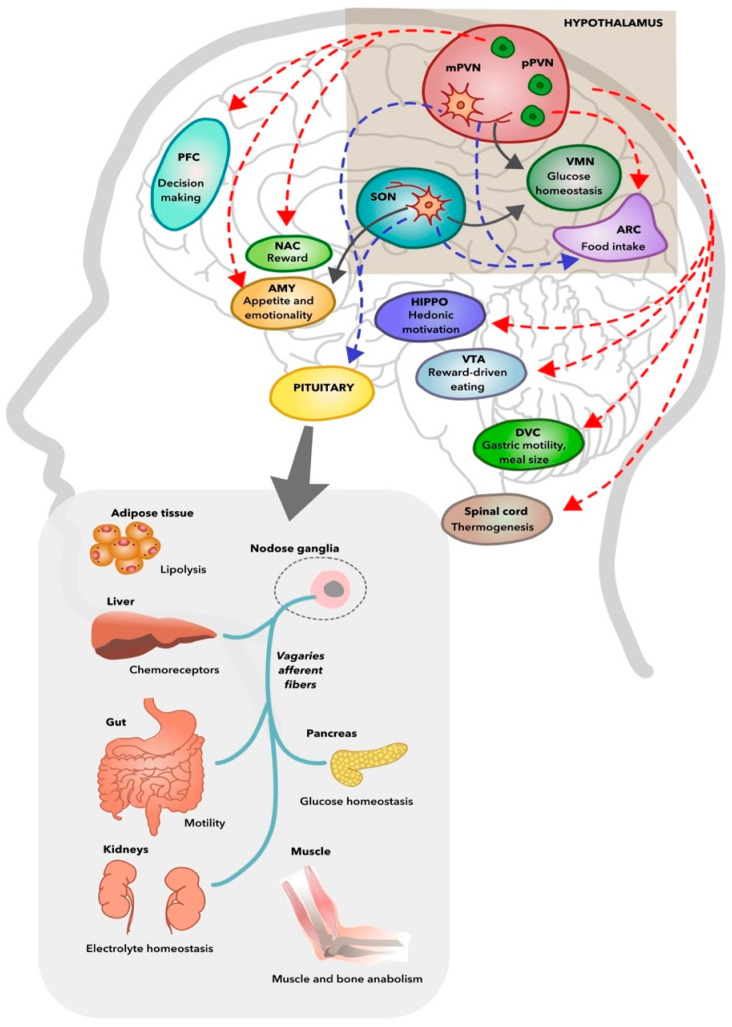
The central and peripheral targets of the brain oxytocin systems in metabolic regulation. Oxytocin is synthesized from the magnocellular neurons of both the paraventricular nucleus (PVN) and supraoptic nucleus (SON). The PVN and SON are sensitive to nutrients (e.g., glucose, sucrose, and leucine) and other hormones (e.g., leptin, insulin, cholecystokinin, and glucagon-like peptide type 1) and influence energy intake and energy balance. Magnocellular oxytocin neurons in the PVN (mPVN) and SON possess axonal projections to the neurohypophysis of the pituitary, from which oxytocin is released into systemic circulation. Parvocellular oxytocin neurons in the PVN (pPVN) project axons to a variety of brain regions, including to the arcuate nucleus (ARC) (receives afferent oxytocin fibers from both mPVN and pPVN), the ventral tegmental area (VTA), the nucleus accumbens (NAc), the amygdala (AMY), the hippocampus (HIPPO), prefrontal cortex (PFC), dorsal vagal complex (DVC), and spinal cord, each of which contains oxytocin receptor-expressing neurons and has important involvement in the regulation of energy balance. Moreover, extrasynaptic oxytocin release from the dendrites of magnocellular oxytocin neurons in the PVN and SON into the ventromedial nucleus (VMN) and AMY are also seen. Oxytocin exerts metabolic effects in multiple organ systems (e.g., gastrointestinal motility, muscle and bone anabolism, lipolysis, and pancreatic insulin secretion) through the secretion of oxytocin in the periphery via the neurohypophysis of the pituitary.

**Table 1 ijms-23-07092-t001:** Clinical operant definitions of metabolic syndrome.

Criteria		WHO (1998)	EGIR (1999)	NCEP: ATP III (2001)	AACE (2003)	IDF (2005)	AHA/NHLBI (2009)
**Central obesity**	WC	-	≥94 cm (M) ≥80 cm (W)	>102 cm (M) >88 cm (W)	-	≥94 cm (M) ≥80 cm (W) *	≥94 cm (M) ≥80 cm (W) *
	BMI	>30 kg/m^2^	-	-	-	>30 kg/m^2^	-
	WTH	>0.90 (M) >0.85 (W)	-	-	-	-	-
**Increased TG**	TG	≥150 mg/dL ^#^	>177 mg/dL or under Tx ^#^	≥150 mg/dL	≥150 mg/dL	≥150 mg/dL or under Tx	≥150 mg/dL or under Tx
**Reduced HDL**	HDL	<35 mg/dL (M) <39 mg/dL (W) ^#^	<39 mg/dL or under Tx ^#^	<40 mg/dL (M) <50 mg/dL (W)	<40 mg/dL (M) <50 mg/dL (W)	<40 mg/dL (M), <50 mg/dL (W), or under Tx	<40 mg/dL (M), <50 mg/dL (W), or under Tx
**Increased BP**	BP	≥160/90 mmHg	≥140/90 mmHg or under Tx	≥130/85 mmHg	>130/85 mmHg	≥130/85 mmHg or under Tx	≥130/85 mmHg or under Tx
**Increased glucose concentration**	IFG	≥110 mg/dL	≥110 mg/dL (2 measures)	≥110 mg/dL	110–125 mg/dL	≥100 mg/dL or under Tx	≥100 mg/dL or under Tx
	IGT ^¶^	140–200 mg/dL	-	-	140–200 mg/dL	-	-
	IR	Yes ^†^	Yes ^‡^	-	-	-	-
	DM	≥126 mg/dL	-	-	-	Yes	Yes
**Other**	MA	UAE ≥ 20 ug/min UACR ≥ 20 mg/g	-	-	-	-	-
**Diagnosis**		IGR ^§^, DM, or IR + ≥2 other criteria	IGR ^§^ or IR + ≥2 other criteria	≥3 criteria	≥2 criteria	Central obesity + ≥2 other criteria	≥3 criteria

^¶^ Impaired glucose tolerance is defined as 2-h glucose levels of 140–200 mg/dL on the 75-g oral glucose tolerance test. ^†^ Insulin resistance refers to glucose uptake below lowest quartile for background population under investigation in euglycemic hyperinsulinemia conditions. ^‡^ Insulin resistance refers to top 25% of fasting insulin concentrations from non-diabetic population. ^§^ Impaired glucose regulation refers to impaired fasting glucose or impaired glucose tolerance. * Central obesity is defined as ethnicity-specific values of waist circumference: United States ≥ 102 cm (men) or ≥88 cm (women); Europids ≥ 94 cm (men) or ≥80 cm (women); and Asians ≥ 90 cm (men) or ≥80 cm (women). ^#^ Increased triglyceride and decreased high-density lipoprotein cholesterol are considered one criterion in WHO (1998) and EGIR (1999) definitions. Abbreviations: M = men; W = women; BMI = body mass index; BP = blood pressure; DM = diabetes mellitus; HDL = high-density lipoprotein cholesterol; TG = triglyceride; IGR = impaired glucose regulation; IFG = impaired fasting glucose; IGT = impaired glucose tolerance; IR = insulin resistance; MA = microalbuminuria; Tx = treatment; UAE = urinary albumin excretion rate; UACR = urinary albumin-to-creatinine ratio; WC = waist circumference; WTH = waist-to-hip ratio; WHO = World Health Organization; EGIR = European Group for the Study of Insulin Resistance; NCEP: ATP III = National Cholesterol Education Program Adult Treatment Panel III; AACE = American Association of Clinical Endocrinology; IDF = International Diabetes Federation; AHA/NHLBI = American Heart Association/National Heart, Lung, and Blood Institute.

## Data Availability

Not applicable.

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
