# Peer review of "Crosstalk between Schizophrenia and Metabolic Syndrome: The Role of Oxytocinergic Dysfunction"

_ijms, 2022, doi:10.3390/ijms23137092_

Round 1

Reviewer 1 Report

In this review, the authors show an overview of the literature on the Crosstalk between Schizophrenia and Metabolic Syndrome with a special focus on the role of oxytocin in these diseases. This new review is particularly important to present the most recent data, as this field of research is generating substantial interest with several new studies being published in the past 2 years The manuscript is written in a comprehensive way, but I considered this paper needs some minor revision:

1. Table 1 needs an improvement in the design because the information is overlapping, so, I suggest establishing a clear division between the different criteria.

2. Deeper discussion about the fact that metabolic syndrome parameters are evaluated in different ways depending on the continent could affect the conclusions of the studies.

3. There are big paragraphs during the manuscript, I suggest separating them into short paragraphs keeping in mind the topic that they are describing.

4. Check some grammar in the manuscript (Example: Lines 360-361 “are has not been”) and clarity in the sentences (Example: Lines 403-405).

5.  Introduction of a new table or figure with the different neurotransmitters and hormones of metabolic syndrome in persons with Schizophrenia is recommendable for a better understanding of the topic.

Author Response

We thank the reviewer for this positive feedback and for the constructive comments on our manuscript. Please see the attachment for our response. 

Reviewer 2 Report

This review addresses an important and interesting topic. Widely it is well written. Nevertheless clarifications and corrections are necessary.

Obviously this is a narrative review. No search strategy is provided by the authors. Please explain how the cited studies were found and selected.

The manuscript  deals with the crosstalk between schizophrenia and metabolic syndrome. For the definition of schizophrenia just one article [1] is provided. Since various studies on „persons with schizophrenia“ or dealing with „schizophrenia“, respectively  are included, it should be clarified whether the diagnostic critieria in all these studies are homogenous, for example all meeting DSM-5 definition of schizophrenia.

The English is widely appropriate. Some errors in syntax and grammar which need correction are described below. An addtional language check is recommended.

The following changes and clarifications are necessary.

Line (l) 49 - please rephrase „overactivation of downregulation“ – „or“ instead of „of“?

l 74 - please errase „in“

l 174 – please change „increase“ to „increased“. Obviously it is suggested that tobacco use contributes to weight gain.  The cited reference [53] deals according to the title with nicotine. It is well established that nicotine induces weight loss. See for example Audrain-McGovern & Benowitz, Clin Pharmacol Ther. 2011; 90: 164-8. Please clarify.

l 192/193 – what is meant with negative effects? Do triglycerides, blood pressure etc. decrease or detoriate after antipsychotics ?

l 270 – 272 - The authors state it would be contradictory that ghrelin levels decreased after treatment with antipsychotics whereas in vitro ghrelin is orexogenic. Please note that indeed ghrelin induces appetite, food intake and weight gain. On the other hand ghrelin plasma levels decline after weight gain. Ghrelin levels are high in anorectic patients and are low in obese subjects. Only in  patients with Prader Willi syndrome obesity and high ghrelin levels coinicide.

l 360/361 – please rephrase „…are has not been…“

l 445-447 – please give references for the roles of oxytocine in love etc. mentionned here.

l 566 – please erase „the“ before „oxytocin“

l 672 – please erase „the“ before „appetite“

l 693 – please change „effect“ to „effects“

l 713 - please change „in the animal study“ to „in an animal study“. Please change „improves“ to „improved“.

l 731 – please rephrase „are further been elucidated“

l 812 – please insert „is“ between „that“ and „common“

l 878 and 881 – „psychosis“ is not specific for schizophrenia, which is probably meant

l 868 to 889 – A subheading like „Conclusions“ would be appropraite for this §

l 889 – The mansucript does not provide much new mechanistic insight particularly into the treatment of schizophrenia as stated here.  

Author Response

(The authors gave the same response as above.)

Reviewer 3 Report

The authors aimed at outlining the crosstalk between schizophrenia and metabolic  syndrome with emphasis on the role of oxytocinergic disfunction. The authors refers to the previous evidences and studies. However I have some suggestions:

1. Overall the review can be much more comprehensive and compact. Elaboration of known facts unnecessarily increases the length and doesn't contribute to any new information. For e.g., Line 436-497 can be eliminated or condensed.

2. There are a number of typos in the beginning of the review, in the abstract (line 18, 41). 

3. It will be great to add author's/current field perspective as well.

Author Response

(The authors gave the same response as above.)

Round 2

Reviewer 2 Report

The manuscript  was clearly improved. The authors have edited it appropriately according to the comments of the reviewers. However, some formulations resulted from the revision, which are in turn in need of correction.

Abstract, lines 24 - 26. It is difficult to understand the sentence "Oxytocin ... association."

line (l) 333 - "is involved" appears twice in one sentence. This is a matter of style. I recommend to rephrase. 

l 1087 and 1088 - I recommend to rephrase "are  associated" in   "persons with...", "those with ..." , respectively

Finally in l 470  - please errase s in "glucocorticoids"

Author Response

We thank the reviewer for this positive feedback and for pointing out the grammar errors. We have revised the sentences accordingly.
